# Bioactive Components and Antioxidant Activity Distribution in Pearling Fractions of Different Greek Barley Cultivars

**DOI:** 10.3390/foods9060783

**Published:** 2020-06-12

**Authors:** Maria Irakli, Athina Lazaridou, Ioannis Mylonas, Costas G. Biliaderis

**Affiliations:** 1Hellenic Agricultural Organization-Demeter, Institute of Plant Breeding and Genetic Resources, P.O. Box 60411, Thermi, 57001 Thessaloniki, Greece; ioanmylonas@yahoo.com; 2Department of Food Science and Technology, School of Agriculture, Aristotle University of Thessaloniki, P.O. Box 235, 54124 Thessaloniki, Greece; athlazar@agro.auth.gr (A.L.); biliader@agro.auth.gr (C.G.B.)

**Keywords:** barley cultivars, pearling fractions, β-glucan, protein, tocols, phenolics, proanthocyanidins, antioxidant activity

## Abstract

In this study, three pearling fractions, namely bran, dehulled grains and pearled grains, derived from fourteen hulled and one hull-less Greek barley cultivars (*Hordeum vulgare* L.), were analyzed for the protein, ash, β-glucan, phenolic compounds and tocols contents. High variations appeared in the bioactive contents across the barley cultivars and fractions as well. The protein and ash contents decreased from the outer to the inner layers, whereas β-glucans presented an inverse trend. The highest protein and β-glucan contents were in the hull-less cultivar; however, one hulled cultivar (Sirios) exhibited similar β-glucan content, while another (Constantinos) had even higher protein content. The results also revealed that functional compounds were mainly located in bran fraction. Similar trends were also noted for the antioxidant activity. Ferulic acid was the primary phenolic acid in all fractions, followed by sinapic and p-coumaric acids that were dominant in bound form. However, oligomeric flavonoids, such as prodelphinidin B_3_, catechin, and procyanidin B_2,_ were more abundant in free form. Overall, this study highlights that different barley cultivars can provide pearling flour fractions of varying composition (nutrients and bioactives), which have the potential to serve as nutritionally valuable ingredients in formulations of cereal-based functional food products.

## 1. Introduction

Barley (*Hordeum vulgare* L.) is one of the oldest cultivated cereal crops and it is used mainly for feed and malt, while food uses constitute <2% of total production. Recently, there has been a renewed interest in the inclusion of barley and its components in foods with nutritional merit in human health. Research has shown the positive effects of human consumption of barley grains in the glycemic index, body weight control and heart diseases due to β-glucan, a soluble dietary fiber, and the additive and synergistic effects of other bioactives, namely tocols and phenolic compounds [1].

Among all cereals, barley is the richest source of tocols, which consist of tocopherols and tocotrienols [2]. Studies have indicated that α-tocotrienol is a more effective antioxidant than α-tocopherol, but it is less bioavailable after oral ingestion, when both ingested [3]. Barley also contains sufficient amounts of phenolic compounds, both in free and bound types. The flavanols, such as catechin, procyanidins and prodelphinidins are the predominant compounds of barley phenolic extract in the free form, while phenolic acids, such as ferulic, sinapic and coumaric acids, mainly exist in bound form [4,5].

Barley is generally classified as hull-less (naked) and hulled. Hulled barley needs to be processed before human consumption, which mainly alter its nutritional and phytochemical profile [6]. The bioactive components in barley grain are unevenly distributed beyond the endosperm, bran and germ fractions. Tocotrienols and phenolic compounds are concentrated in the external parts of the grain, tocopherols especially are mainly found in the germ, whereas the β-glucan is highly located in inner layers [7,8]. Thus, barley fractionation through milling or abrasion can produce fractions rich in healthy bioactives [9].

Various authors have examined the distribution of bioactive compounds in different milled barley fractions [7,10]. Moreover, earlier studies have investigated the effect of grain type, variety and environment in the allocation of bioactives in the barley grain [2,9,11,12]. Selecting high-quality barley varieties, enriched in bioactive components, is very important for achieving the desired end-product nutritional characteristics.

In Greece, approximately, 400 million tons of hulled barley are produced annually; it is used, in majority, for feed and malt. However, as barley is rich in valuable functional components, such as β-glucan, tocotrienols and phenolic antioxidants, it can be accordingly processed in order to be suitable for human consumption. Even though barley has been widely appreciated for its nutritional potential, there was no published data about the phytochemical profile of endogenous Greek barley cultivars. Therefore, the present work aimed to investigate the distribution and quantification of nutritional and bioactive components in pearled fractions of fifteen Greek barley cultivars. A secondary objective was to evaluate the antioxidant activity of their fractions in order to serve as functional ingredients of functional foods sector.

## 2. Materials and Methods

### 2.1. Barley Samples and Chemicals

Fourteen hulled and one hull-less barley commercial cultivars (*Hordeum vulgare* L.) were cultivated on Thermi’s field area (Thessaloniki, northern Greece, latitude 40°32′49.63″ N, longitude 23°01′10.81″ E) during the 2016–2017 growing season, based on a common experimental design to minimize environmental variations. The following two-rowed hulled barley cultivars were used: Andromeda (ANDA), Niki (NIKI), Persefoni (PENI), Dimitra (DIMA), Constantinos (CONS), Makedonia (MAKA), Thessaloniki (THKI), Sirios (SIOS), Kos (KOS), Thermi (THMI), Bizantio (BIZA), Cyprus (CYPS) and Ippolytos (IPOS). One six-rowed hulled, namely Triptolemos (TIOS), and one hull-less cultivar (HLES) were also included. The mean moisture content of barley grains was approximately 12%. Hulled barley samples (200 g in duplicate) were pearled with a small-scale debranning machine (Satake Engineering, Tokyo, Japan) in order to obtain three fractions: dehulled grains (DG), pearled grains (PG) and bran. The process was monitored by time control. Barley grains were dehulled for different times in order to remove approximately 12% of the initial grain weight by isolating the hull. After this step, the DG samples were further pearled for different times in order to eliminate 25% of the initial grain weight after removal of bran. The residual 75% of the grain and the bran were also collected. For each hulled barley cultivar, a different pearling time was applied. The DG and PG samples were finally milled (ZM-100; Retsch, Haan, Germany) in order to obtain dehulled flour (DF) and pearled flour (PF). All other chemicals and solvents used were of analytical or HPLC grade.

### 2.2. Grain Characteristics

Barley grain weight and dimensions (length, width, thickness) were evaluated using a SeedCount image analysis system, model SC4 (Seed Count, Condell Park, Australia), according to the manufacturer’s instructions. The whiteness and color of PG were also determined by a HunterLab colorimeter, model MiniScan XE Plus (Reston, Virginia, USA) under the CIE L*, a* and b* color values. Each sample was measured five times and the whiteness index (WI) was calculated according to the Hunter whiteness formula: WI = 100 − {(100 − L*)^2^ + a*^2^ + b*^2^}^1/2^.

### 2.3. Chemical Analyses

Compositional analyses of DF and PF, as well as barley bran samples, were measured based on official methods [13]. Crude protein content was determined by the Kjeldahl method, whereas crude ash was determined by dry ashing procedure. The enzymatic determination of β-glucan content was carried out according to the procedure provided by Megazyme International Ireland Ltd. (Wicklow, Ireland) with corresponding test kit.

### 2.4. Phenolics Extraction

A total of 0.4 g barley fractions were extracted two times with 4 mL of 70% aqueous methanol in an ultrasound bath (Thermo Fisher Scientific, Loughborough, England) for 10 min at room temperature, according to the procedure of Irakli et al. [14] with some modifications. Then, after centrifugation at 10,000× *g* for 10 min at 4 °C, the combined extracts were constituted the free phenolics. The residue collected after the free phenolics extraction was subjected to alkaline hydrolysis by adding 20 mL of 4 N NaOH, in order to obtain the bound extracts. The samples were subjected to sonication for 90 min and then centrifugation at 1000× *g* for 10 min; the supernatant was acidified to pH 2.0 with concentrated HCl, and then extracted with 3 × 10 mL ethyl acetate. The organic layers were mixed and evaporated to dryness with the aid of vacuum evaporator at 40 °C and the residue was re-dissolved in 2 mL of the methanol/water (50:50, *v*/*v*). All extracts were stored at −25 °C until analysis.

### 2.5. Total Phenolic Content (TPC)

The TPC of free and bound fractions were determined according to the Folin–Ciocalteu method [15]. Briefly, extracts of 0.2 mL were mixed with 0.8 mL diluted Folin–Ciocalteu reagent (dilution 1:10, *v*/*v* with water), 2 mL sodium carbonate (7.5% *w*/*v*) solution and distilled water until the final volume of 10 mL. The absorbance of mixture was recorded at 725 nm after 60 min incubation in a dark place. The results were expressed as mg of gallic acid equivalents (GAE) per g of sample.

### 2.6. Total Flavonoid Content (TFC)

The TFC was determined using the colorimetric method of aluminum chloride as described by Bao et al. [16] with minor modification. Moreover, 0.2 mL extracts were mixed with 0.15 mL 5% NaNO_2_, 0.15 mL 10% AlCl_3_·6H_2_O and 0.5 mL 1M NaOH; the absorbance was recorder at 510 nm after 30 min incubation. The TFC was expressed as mg catechin equivalent (CATE) per g of sample.

### 2.7. Total Proanthocyanidins Content (TPAC)

TPAC was measured according to butanol-acid assay [17] as follows: 0.5 mL diluted phenolic extract (1:5, *v*/*v*) were mixed with 3 mL of the reagent consisted of n-butanol/HCl (95:5, *v*/*v*) and 0.1 mL of ferric ammonium sulfate (2% *v*/*v* in 2 M HCl). The absorbance of boiled mixtures for 60 min was recorded at 550 nm after cooling. The absorbance of the unheated tubes was considered as blank. TPAC was expressed as mg of procyanidin B_2_ equivalents (mg PCNE) per g sample.

### 2.8. ABTS Radical Scavenging Activity

Radical scavenging activity of barley extracts against ABTS radical cation was determined according to Re et al. [18]. Briefly, ABTS^+^ solution was prepared by mixing 7.4 mM ABTS and 2.6 mM potassium persulfate in equal volumes and adjusted its absorbance of 0.70 ± 0.02 at 734 nm. 3.9 mL of the above ABTS+ solution was added to 0.1 mL of phenolic extract and the absorbance at 734 nm was recorded after 4 min against a blank. Inhibition of ABTS radical cation (%) was calculated according to equation: inhibition (%) = [(Ao − As)/Ao] × 100, where Ao is the absorbance of the blank and As is the absorbance of the sample after 4 min. The results were expressed as mg Trolox equivalents (TE) per g of sample.

### 2.9. Ferric Reducing Antioxidant Power (FRAP)

The reducing power of barley extracts was evaluated based on the method of Benzie and Strain [19]; 0.1 mL of phenolic extract was mixed for exactly 4 min with 3 mL of FRAP solution at 37 °C. The absorbance at 593 nm of the colored product was measured against blank and the results were expressed as mg TE per g of sample.

### 2.10. HPLC Analysis of Phenolic Compounds

Phenolic extracts from both barley free and bound fractions were filtered (pore size 0.2 μm) and analyzed using an HPLC system (Agilent Technologies, 1200 series, Urdorf, Switzerland), accompanied with a Nucleosil 100 C_18_ column (250 mm × 4.6 mm, i.d. 5 μm) and applied the chromatographic conditions, as mentioned by Skendi et al. [20]. The diode array detector recorded the spectra at 280 nm (proanthocyanidins) and 320 nm (phenolic acids). Quantification of phenolic compounds was based on external standards calibration, except for prodelphinidin B_3_ and procyanidin B_3_, which were expressed in catechin equivalents; the results were reported as mg per 100 g of sample.

### 2.11. Tocopherols and Tocotrienols Analysis

A total of 0.25 g of barley flour was mixed with 4 mL hexane for 15 min under sonication and the extract was collected after centrifugation at 1500× *g* for 10 min. The above procedure was repeated one more time. The combined supernatants were evaporated to dryness under the flow of nitrogen, the remaining residue was reconstituted in 400 μL of mixture acetonitrile/methanol (85:15, *v*/*v*) and finally aliquot of 20 μL was injected into the previous HPLC system. The applied chromatographic conditions were described by Irakli et al. [21]. External calibration curves were made based on standard solutions and the results were expressed as μg per g of sample.

### 2.12. Statistical Analysis

All the characterization measures were performed four-fold. All parameters were subjected to one-way analysis of variance (ANOVA) and the Tukey’s test was used to find significant differences among the means. The probability level of 0.05 (*p <* 0.05) was used as a baseline for significance. All statistical analyses were performed by using the Minitab 17 (Minitab Inc., State College, PA, USA) software.

## 3. Results and Discussion

### 3.1. Barley Grain Characteristics

The physical parameters of grains, such as 1000-grain weight (TGW), length, width and thickness, were compared among the various barley cultivars (Table 1). ANOVA indicated significant differences in all studied grain characteristics *(p <* 0.05). TGW varied between 27.8 (IPOS) and 46.7 g (NIKI), with a mean value of 40.6 g. The hulled cultivars had grain length, width and thickness ranging between 7.72 (IPOS) and 10.13 mm (TIOS), 3.06 (IPOS) to 3.82 mm (NIKI) and 2.64 (IPOS) to 2.99 (NIKI and TIOS), with mean values of 8.45, 3.47 and 2.86 mm, respectively. The hull-less cultivar had similar weight (45.2 g), length (8.36 mm), width (3.39 mm) and thickness values (2.78 mm) compared to the hulled cultivars. The obtained grain size and shape values were within the ranges previously reported in the literature [22,23].

The barley pearling process revealed that the time required to remove most of the hulls varied significantly among the hulled cultivars; i.e., the dehulling time ranged from 1.5 (ANDA, NIKI, THKI) to 3.0 min (CONS, CYPS, IPOS) (Figure 1a). The hull yields obtained varied from 12.0 (CONS, MAKA, THKI, SIOS) to 16.0% (TIOS), with a mean value of 12.7% (Figure 1b), depending on the grain size and shape. It was also noticed that TIOS, a 6-rowed barley cultivar, having the highest grain length among all hulled cultivars, gave the highest hull yield.

Further pearling in order to remove bran resulted in bran yields which ranged from 10.1 (TIOS) to 13.9% (DIMA), with a mean value of 12.8%, whereas the debranning time varied from 5.0 (ANDA) to 12.7 min (IPOS) (Figure 1a,b); IPOS had the longest debranning time and the lowest size among all hulled cultivars. This suggests that smaller grains require higher pearling times, probably due to different ratios between width and length of barley grain, which is in accordance with the results of Edney et al. [24]. The hull-less grains appeared the shortest pearling time (5.7 min) along with other hulled barley cultivars, namely ANDA, NIKI, PENI, DIMA and THKI, although their grain characteristics differed significantly (Table 1). In total, hull and bran removal of different barley cultivars were achieved at different pearling times, ranging from 6.5 (ANDA) to 15.5 min (IPOS), to yield residual pearled material of ~75% of the initial grain weight. Significant differences in whiteness and brightness (L*) of the pearled barleys were observed among cultivars. The hull-less barley tended to be significantly whiter than most of the hulled cultivars; for the latter ones, DIMA was the brightest preparation.

Correlation coefficients indicated that pearling time of hulled barley grains was most affected by weight, kernel width and thickness (Table 2). It could be concluded that the smaller, narrower and thinner grains require longer pearling times using the laboratory pearling device, whereas the length had no significant effect. In this study, we have found that the whiteness and brightness of PG were insignificantly correlated (*p* > 0.05) with pearling time, in contrast to the observations of Edney et al. [24], who indicated a significantly negative correlation between them. The whiteness of PG significantly increased with wider, larger and brighter grains, whereas the brightness was indicated highly positive correlation (*p* < 0.001) to grain weight, thickness and width. Positive highly significant correlations were also observed between barley grain dimensions and weight, as well as between grain thickness and width. Correlation analysis showed that grain weight and size are important parameters that should be considered in the barley pearling process.

### 3.2. Chemical Characterization of Barley Pearling Fractions

Protein and ash contents are the most important qualitative traits in the milling industry; moreover, in barley processing, β-glucan content must be also taken into consideration in view of its positive impact on human health [11]. Chemical composition results of the three barley fractions indicated that protein and ash contents increased more from the endosperm fractions (DF and PF) towards to bran, while β-glucan content followed the reverse trend (Table 3). The protein content varied from 9.60 to 16.06% and from 8.51 to 14.58%, with mean values of 13.58% and 12.03%, respectively, among DF and PF fractions. CONS had the highest and DIMA the lowest protein content in both fractions among barley cultivars. Ash in DF and PF fractions varied from 1.22 to 1.89% and from 0.54 to 1.21%, with mean values of 1.57% and 1.95%, respectively; IPOS, CYPS and KOS exhibited the highest ash content, whereas DIMA exhibited the lowest in both fractions. As the bran mainly includes the external layers of the dehulled grain, it had the highest content of ash and protein. Among all the bran preparations, the CONS sample had the highest protein content, while that of HLES was the richest source of minerals.

β-glucan contents were significantly differed between barley fractions in all cultivars and the trends for most of the cultivars followed the decreasing order of: PF > DF > bran (with mean values of 4.53, 4.27 and 3.71%, respectively). However, in THKI, barley cultivar, the β-glucan content decreased in the PF instead of DF. It has been noticed that the distribution of β-glucan varies with barley cultivars, in accordance with other studies [6,25,26]. Concentration of β-glucan for PF varied from 3.75 to 5.27%, for DF from 3.41 to 5.25% and for bran from 2.98 to 4.78% (Table 3). It is worth noticing that as the external barley layers are removed during the pearling process, the β-glucan content increased, in accordance with previous findings [2]. This could justify why pearling of 25% of the grain had small effect on β-glucan content. However, Zheng et al. [27] found that the flour produced after roller-milled of barley grains had the lowest concentration of β-glucan, in contrast to bran. Among all studied hulled cultivars, SIOS had the greatest β-glucan content in all fractions (mean values of 5.18, 5.08 and 4.78% for PF, DF and bran, respectively), whereas the THMI had the lowest ones among all samples. The superior barley cultivar, SIOS, among other barley cultivars may be due to the genotype effect, since all the cultivars were grown under the same conditions. The importance of genetic factors has also been reported by several researchers [2,28]. As it was expected, the HLES exhibited the highest β-glucan level in the PF and DF fractions; instead, its bran fraction had the lowest β-glucan content, along with the THMI. Similarly, Izydorczyk et al. [25] also found that barley pearling by-products were contained high levels in arabinoxylans, protein and ash and low amounts of β-glucans and starch. On the contrary, in two hull-less and two waxy barleys, the greatest β-glucan content was determined in the bran, according to Šimic et al. [29].

### 3.3. Phenolic Contents and Antioxidant Activity of Barley Pearling Fractions

In our study, TPC in barley fractions were determined in free and bound extracts, and the results are presented in Figure 2a. Generally, TPC significantly varied within cultivars and pearling fractions, and the distribution followed a decreasing order of: bran > DF > PF. The TPC of DF and PF extracts ranged from 1.56 to 2.45 mg GAE/g and from 1.08 to 1.73 mg GAE/g, with mean values of 1.95 and 1.31 mg GAE/g, respectively. Among all barley cultivars, PENI had the highest values and the THMI the lowest ones. The TPC of HLES did not differ from those of hulled cultivars, which is in accordance with an earlier report [10].

The highest amount of TPC in bran fraction was found in TIOS and HLES (6.05 and 5.98 mg GAE/g) and the lowest in THMI (3.46 mg GAE/g), while the mean value among all barley cultivars was 4.63 mg GAE/g. Similarly, Šimic et al. [29] reported that TPC of a hull-less cultivar was greatest in the bran in comparison to the refined flour; however, they found lower values for bran than those from our study. On the other hand, Moza and Gujral [30] reported similar values of TPC in the bran fraction in hull-less cultivars (3.67–4.44 mg FAE/g). Therefore, selection of suitable cultivar for pearling is fundamental factor to take barley flour fractions with different amounts of phenolics.

In the three fractions, the TFC in barley cultivars were evaluated, and there were significant differences among them (*p* < 0.05) (Figure 2b). The TFC of DF and PF extracts ranged from 0.54 to 0.98 mg CATE/g and from 0.28 to 0.54 mg CATE/g, with mean values of 0.73 and 0.45 mg CATE/g, respectively. Among the tested barley cultivars, KOS had the highest TFC and CONS the lowest one in DF, while for the PF fraction, IPOS and HLES showed the greatest TFC values and CONS the lowest one. The bran fraction contained on average 3.0 and 4.6 times greater concentrations of TFC than the DF and PF, respectively. This indicates that flavonoids are found mainly in the external layers of the grain compared to the endosperm, which is consistent with the results of Gangopadhyaya et al. [8].

Proanthocyanidins, the major types of flavonoids in barley grain, are oligomeric and polymeric flavan-3-ols that exert strong antioxidant activity and other related health benefits. TPAC were quantified in free extracts, because the contribution of bound extracts was very low. The TPAC in DF and PF fractions ranged from 0.23 to 0.92 mg PCNE/g and from 0.17 to 0.63 mg PCNE/g, with mean values of 0.56 and 0.33 mg PCNE/g, respectively (Figure 2c). For the DF fractions, IPOS had the highest TPAC, while HLES the lowest concentration among the barley cultivars, whereas among the PF fractions, BIZA had the greatest TPAC and CONS the lowest one. Dvorakova et al. [31] also reported similar TPAC contents, ranging from 0.89 to 2.00 mg CE/g for ten barley varieties. However, Kim et al. [32], found much lower TPAC values (55.3–83.0 μg of catechin/g), by applying the vanillin test in seven pigmented barley varieties. Although some studies have shown higher TPAC in hull-less compared to hulled barley [32,33], in our study the opposite was observed with the hull-less barley compared to all other hulled cultivars.

For the hulled cultivars, significant differences were evident among the three flour fractions with the TPAC content showing the order: bran > DF > PF. The bran fraction appeared to have the highest TPAC concentration (0.97–2.47 mg PCNE/g) among others, with a mean value of 1.72 mg PCNE/g; the greatest level occurred in TIOS, whereas the lowest in CONS.

### 3.4. Quantification of Flavanols and Phenolic Acids in Barley Pearling Fractions

As the butanol-acid assay (colorimetric test) for quantifying TPAC in barley is not specific for total proanthocyanidins, the HPLC analysis was applied for quantifying the major oligo- or polymer of monomeric flavanols (proanthocyanidins derivatives) as well as the phenolic acids (PAs). Flavanol levels determined from the fifteen barley cultivars were based on the four main compounds, namely prodelphinidin B_3_ (PDB_3_), procyanidin B_2_ (PCB_2_), procyanidin B_3_ (PCB_3_) and catechin (CAT) that were identified and quantified by chromatography. The flavanols levels ranged from 93.2 to 214.0 mg/100 g, from 39.2 to 67.2 mg/100 g and from 37.2 to 74.5 mg/100 g for the bran, DF and PF fractions, with mean values of 149.4, 55.2 and 53.1 mg/100 g, respectively (Table 4). As was expected, the flavanols were mainly concentrated in bran and their content was about 3 times higher than in DF and PF, whereas between PF and DF there were no significant differences. In the current study, PENI was found to contain the highest amount of total flavanols compared to other cultivars. Similarly, Yoshida et al. [34] reported TPAC of hulled and hull-less barley grown in Japan ranging from 0.12 to 0.80 mg/g. Kim et al. [32] instead, reported lower TPAC levels in pigmented barleys from Korea, ranging from 0.02 to 0.13 mg/g. The wide variation in TPAC could be due to genetic factors and the extraction process conditions employed.

As it is shown in Figure 3a, PDB_3_ was the predominant flavanol in all fractions, being present at the highest concentrations in the bran fractions. As the number of barley cultivars is large, only the most representative barley cultivars, in respect to high and low levels of TPAC, are presented herein. The levels of PCB_3_, PCB_2_ and CAT in PF and DF fractions were dependent on the barley cultivar. In general, the level of PCB_3_ was higher in PF than DF fraction, whereas CAT levels did not seem to vary among the fractions of most cultivars tested.

Five PAs, namely vanillic acid (VA), caffeic acid (CA), p-coumaric acid (pCA), ferulic acid (FA) and sinapic acid (SA), were also quantified in both the free and bound phenolic extracts, and the total (sum of free and bound) are summarized in Table 3. The major PA in all barley fractions was FA, followed by SA and pCA in the extracts of bound phenolics; the contribution of FA to total PAs was more than 93% (data not shown). In general, the barleys contained good levels of total PAs, although significant differences were observed among the various cultivars and their fractions as well. Other investigations have indicated that the predominant phenolic compounds in barley are PAs that are mainly located in external layers of the grain [12]. Similarly, in our study, we found that the bran fraction had the greatest total PAs content, followed by DF and PF, with mean values of 137.5, 53.4 and 25.0 mg/100 g, respectively. In particular, bran derived from TIOS and IPOS seem to have the highest total PAs, while the DF and PF flour fractions of THKI had the highest content of PAs. As expected, FA was the main phenolic acid in all fractions among the barley cultivars (Figure 3b), in accordance with previous findings [32]. FA accounted for 50–65% of the total amount of PAs, followed by SA (20–30%) and pCA (3–8%) in all barley fractions.

### 3.5. Tocopherols and Tocotrienols in Barley Pearling Fractions

The mean values of tocol content in bran, DF and PF fractions for all barley cultivars was 186.0, 53.0 and 29.4 μg/g, respectively (Table 4). There was a considerable variation of tocol contents among cultivars and fractions. The results demonstrate that tocol content varies with cultivar; the highest tocol content in DF was for IPOS (71.7 μg/g), whereas the lowest for HLES (36.3 μg/g). The PF fractions were generally found to have two times lower tocol content than their DF counterparts, whereas the bran fractions were the richest in tocol content among all fractions. Comparing the different bran fractions, IPOS, TIOS, THKI and DIMA were the best sources of tocols, with a mean value of 235.9 μg/g, while for the DF fractions IPOS, THMI and PENI was the richest sources of these compounds, with a mean value of 68.5 μg/g. The HLES cultivar showed relatively low value of total tocols in all fractions, in accordance with other studies [28].

In particular, tocotrienols in bran fractions accounted for about 86% of total tocols, whereas the corresponding value for DF and PF was 80%, which means that tocotrienols are mainly located in the aleurone layer, as earlier found by other authors [2]. Moreover, α-tocotrienol was the main homologue accounting for about 56% of total tocols and for 50% of total tocotrienols (data not shown). These results are in good agreement with those of Panfili et al. [2]. High contents were also found for α-tocopherol, accounting for about 10%–33% of total tocols, for γ-tocotrienol (~15–25%) and for β-tocotrienol (~5–25%).

### 3.6. Antioxidant Activity of Barley Pearling Fractions

ABTS and FRAP values varied significantly among cultivars and barley fractions (Figure 4). Generally, a decrease in antioxidant activity of the barley fractions appeared upon pearling. This is in agreement with other studies [11], where a decrease in antioxidant activity followed the trend: bran > DF > PF. The ABTS values of DF and PF fractions were in the range of 3.16–4.52 mg TE/g and 1.67–3.45 mg TE/g, with mean values of 3.71 and 2.36 mg TE/g, respectively (Figure 4a). The total ABTS value of the bran was the highest among the three fractions, ranging from 8.53 to 15.68 mg TE/g, with a mean value of 12.49 mg TE/g. The bran preparations of PENI, TIOS and HLES cultivars exhibited the highest total ABTS values, whereas for the PF and DF fractions the PENI cultivar exhibited the highest value.

The FRAP values of the barley fractions among cultivars, are also presented in Figure 4b. The FRAP values for DF and PF fractions were in the range of 1.94–3.60 mg TE/g and 0.83–1.96 mg TE/g, with mean values of 2.81 and 1.26 mg TE/g, respectively. The total FRAP values of the bran samples were highest among all fractions, ranging between 8.12 and 14.49 mg TE/g, with a mean value of 10.99 mg TE/g. Similar trends to those noted for ABTS among the flour preparations of all cultivars were also observed for the FRAP values.

The extracts of free phenolics had higher total ABTS values than the extracts of bound phenolics, accounting from 53 to 78% (mean value of 66%), from 59 to 77% (68%) and from 61 to 78% (69%) to total ABTS value for bran, DF and PF fractions, respectively (data not shown). On the other hand, the extracts of free phenolics (53, 51 and 56%) and bound phenolics (47, 49 and 44%) contributed almost equally to the FRAP capacity value for the bran, DF and PF fractions, respectively. Similar results were reported by Yang et al. [35] studying twelve barley varieties.

### 3.7. Correlation Analysis

Pearson correlations between antioxidant activity (ABTS and FRAP values) and major phytochemical components (TPC, TFC, TPAC, Flavanols and Tocols) are shown in Table 5. In general, highly significant correlations (*p* < 0.001, *n* = 45) were observed between all the phytochemical compounds analyzed, presenting correlation coefficients from 0.88 to 0.99.

The highest positive correlations were between the antioxidant capacities evaluated by the two assays and the TPC (ABTS: *r* = 0.98, *p* < 0.001; FRAP: *r* = 0.98, *p* < 0.001), whereas the lowest were between FRAP and TPAC. This clearly indicates that phenolic compounds contribute mainly to antioxidant activities of the barley flour fractions. The highest Pearson’s coefficient was obtained between the ABTS and FRAP antioxidant data (*r* = 0.99, *p* < 0.001), probably due to the same reaction mechanism the two assays have. Moreover, there were highly significant positive correlations between the tocols contents and the antioxidant activity values as assessed by the two methods.

## 4. Conclusions

This study demonstrated that pearling can be an effective milling approach to modulate the distribution of nutrients and health promoting phytochemicals in the barley flour fractions for specific end-use. The results indicated that the majority of nutritional and bioactive components exhibit a decreasing concentration from the surface layers to the center part of the grain among all barley cultivars tested, with the exception of β-glucan, which is mostly located in the endosperm. Significant differences in the contents of proteins, ash, β-glucans, phenolic acids, proanthocyanidins, tocotrienols and tocopherols, as well as in antioxidant capacity, were found among the fifteen barley cultivars tested. The large variations noted in composition among the various flour fractions from different cultivars, most likely reflecting factors related to genetic diversity, point to the need for proper selection of the raw grain material and the appropriate processing scheme to generate flour streams suitable for specific applications in the food, feed and other industrial sectors. Overall, this study provides the foundation for promoting the cultivation of endogenous barley cultivars with the aim to find alternative uses for barley flour fractions enriched in health promoting components and, thereby, assisting the enlargement of the cereal-based functional foods market.

## Figures and Tables

**Figure 1 foods-09-00783-f001:**
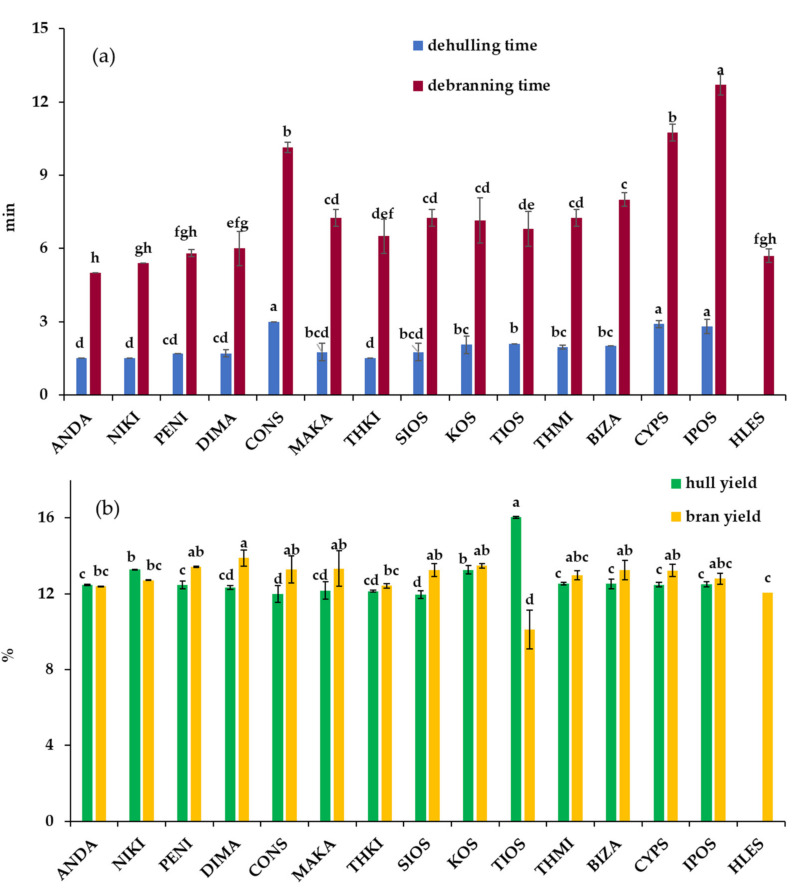
Comparison of dehulling and debranning time (**a**) and hull and bran yield (**b**) among barley varieties tested. Bars with no letter in common are significantly different from each other (*p* ≤ 0.05). Correlation coefficients indicated that pearling time of hulled barley grains was most affected by weight, kernel width and thickness (Table 2). It could be concluded that the smaller, narrower and thinner grains require longer pearling times using the laboratory pearling device, whereas the length had no significant effect. In this study, we have found that the whiteness and brightness of PG were insignificantly correlated (*p* > 0.05) with pearling time, in contrast to the observations of Edney et al. [24], who indicated a significantly negative correlation between them. The whiteness of PG significantly increased with wider, larger and brighter grains, whereas the brightness was indicated highly positive correlation (*p* ≤ 0.001) to grain weight, thickness and width. Positive highly significant correlations were also observed between barley grain dimensions and weight, as well as between grain thickness and width. Correlation analysis showed that grain weight and size are important parameters that should be considered in the barley pearling process.

**Figure 2 foods-09-00783-f002:**
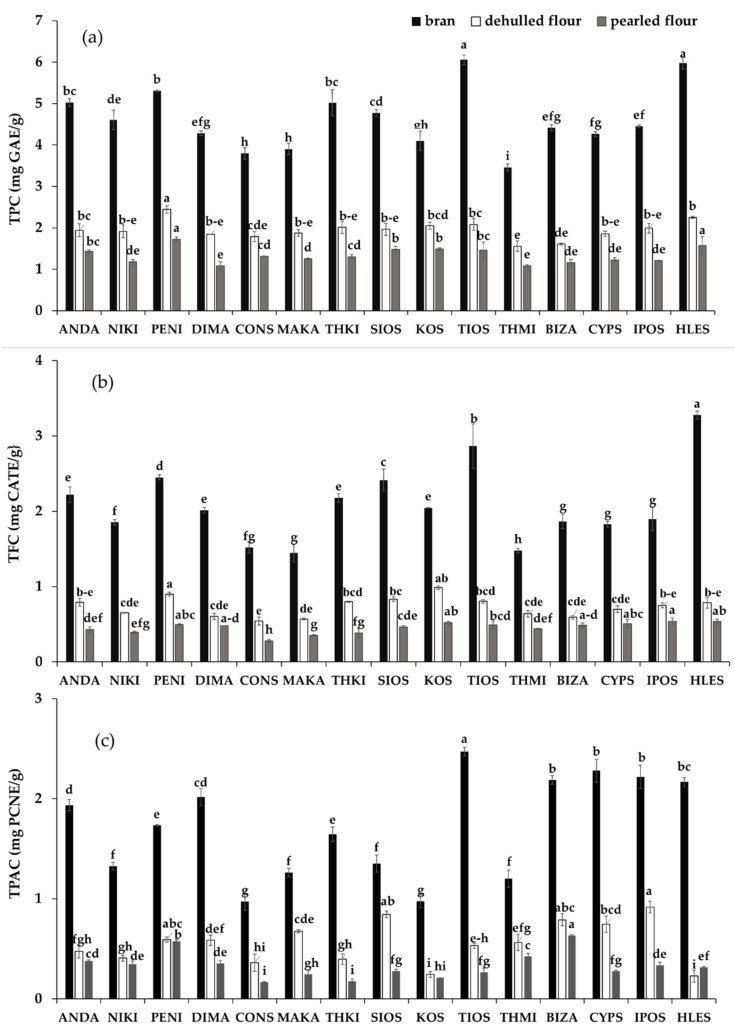
Total phenol content (**a**), total flavonoid content (**b**), total proanthocyanidins content (**c**) of fractions (bran, dehulled four, pearled flour) of barley cultivars. Bars with no letter in common are significantly different from each other (*p* ≤ 0.05).

**Figure 3 foods-09-00783-f003:**
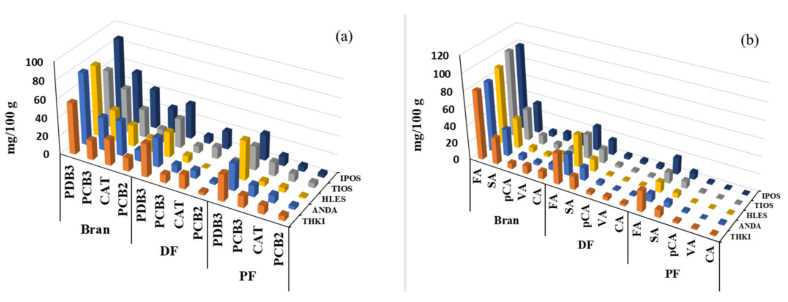
Individual flavanols (**a**) and phenolic acids (**b**) of fractions of selected barley cultivars. THKI, thessaloniki; ANDA, andromeda; TIOS, triptolemos; IPOS, ippolytos; HLES, hull-less; PDB3, prodelphinidin B3; PCB3, procyanidin B3; PCB2, procyanidin B2; CAT, catechin; FA, ferulic acid; SA, sinapic acid; pCA, p-coumaric acid; VA, vanillic acid; CA, caffeic acid; DF, dehulled flour; PF, pearled flour.

**Figure 4 foods-09-00783-f004:**
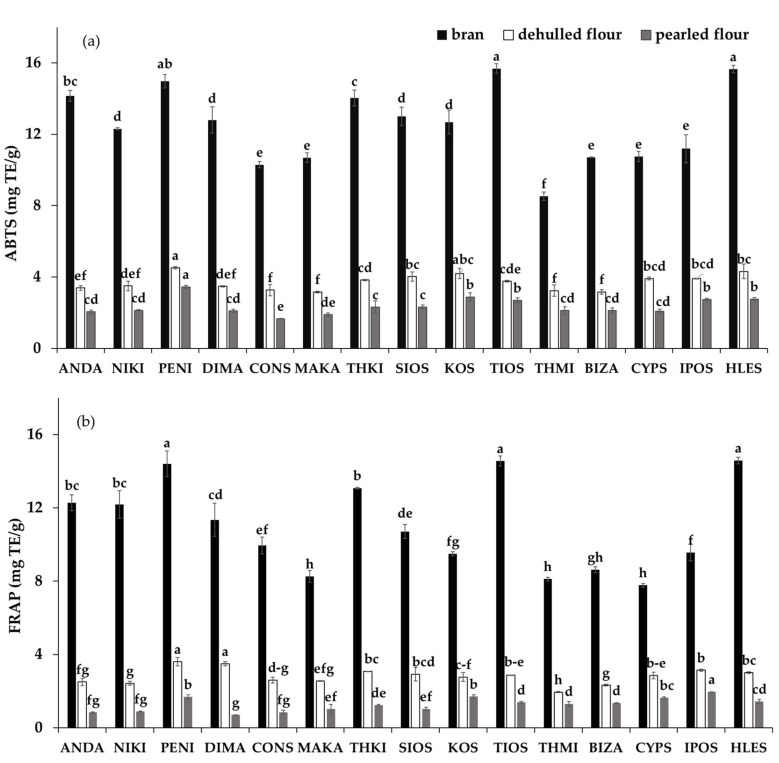
Antioxidant activity by ABTS (**a**) and FRAP tests (**b**) of fractions (bran, dehulled four, pearled flour) of barley cultivars. Bars with no letter in common are significantly different from each other.

**Table 1 foods-09-00783-t001:** Mean physical characteristics of the commercial barley cultivars used in the present study.

Cultivar	TGW (g)	Length (mm)	Width (mm)	Thickness (mm)	Pearling Time (min)	PG Whiteness	PG L*
ANDA	43.5 ^c^	8.57 ^cd^	3.58 ^cd^	2.93 ^ab^	6.5 ^fg^	38.8 ^e^	91.2 ^ab^
NIKI	46.7 ^a^	8.25 ^fg^	3.82 ^a^	2.99 ^a^	6.9 ^ef^	44.0 ^cd^	91.0 ^abc^
PENI	37.0 ^f^	8.20 ^g^	3.36 ^fg^	2.92 ^ab^	7.5 ^ef^	21.4 ^j^	90.5 ^de^
DIMA	43.2 ^c^	8.05 ^h^	3.78 ^a^	2.92 ^ab^	7.7 ^e^	45.3 ^bc^	91.4 ^a^
CONS	37.0 ^f^	7.90 ^i^	3.45 ^e^	2.74 ^cd^	13.2 ^b^	38.8 ^e^	90.1 ^ef^
MAKA	43.1 ^c^	8.50 ^cde^	3.60 ^c^	2.91 ^ab^	9.0 ^c^	36.6 ^fg^	89.5 ^g^
THKI	40.2 ^e^	8.60 ^c^	3.42 ^ef^	2.86 ^bc^	8.0 ^de^	26.8 ^i^	89.4 ^g^
SIOS	45.5 ^ab^	8.58 ^cd^	3.66 ^b^	2.92 ^ab^	9.0 ^cd^	46.3 ^ab^	90.7 ^cd^
KOS	35.6 ^g^	8.19 ^gh^	3.34 ^g^	2.75 ^cd^	9.2 ^c^	31.1 ^h^	88.4 ^h^
TIOS	41.8 ^d^	10.13 ^a^	3.25 ^h^	2.99 ^a^	8.9 ^cd^	35.6 ^g^	90.3 ^def^
THMI	43.1 ^cd^	8.22 ^fg^	3.52 ^d^	2.91 ^ab^	9.2 ^c^	35.9 ^g^	90.5 ^de^
BIZA	45.2 ^b^	9.08 ^b^	3.59 ^c^	2.92 ^ab^	10.0 ^c^	47.8 ^a^	90.9 ^bc^
CYPS	36.6 ^fg^	8.45 ^de^	3.26 ^h^	2.77 ^c^	13.7 ^b^	43.0 ^d^	90.7 ^cd^
IPOS	27.8 ^h^	7.72 ^j^	3.06 ^i^	2.64 ^d^	15.5 ^a^	38.4 ^ef^	89.4 ^g^
HLES	45.2 ^b^	8.36 ^ef^	3.39 ^fg^	2.78 ^c^	5.7 ^g^	48.0 ^a^	89.2 ^g^
*Mean*	*40.6*	*8.45*	*3.47*	*2.86*	*9.3*	*38.5*	*90.3*
*CV%*	*12.0*	*6.6*	*5.8*	*3.6*	*29.1*	*19.7*	*0.9*

Means in each column followed by a common letter are not statistically different (*p* > 0.05). TGW: 1000-grain weight; PG: pearled grains; L*: brightness; CV: coefficient of variance. ANDA, andromeda; NIKI, niki; PENI, persefoni; DIMA, Dimitra; CONS, constantinos; MAKA, makedonia; THKI, Thessaloniki; SIOS, sirios; KOS, kos; TIOS, triptolemos; THMI, thermi; BIZA, bizantio; CYPS, cyprus; IPOS, ippolytos; HLES, hull-less. (ANDA, andromeda; NIKI, niki; PENI, persefoni; DIMA, Dimitra; CONS, constantinos; MAKA, makedonia; THKI, Thessaloniki; SIOS, sirios; KOS, kos; TIOS, triptolemos; THMI, thermi; BIZA, bizantio; CYPS, cyprus; IPOS, ippolytos; HLES, hull-less).

**Table 2 foods-09-00783-t002:** Correlation coefficients among grain characteristics used to predict its effect on barley pearling.

	Pearling Time	Pearled Whiteness	Pearled Brightness	Weight	Length	Thick-ness	Width
Pearling time	1.00						
Pearled whiteness	0.05	1.00					
Pearled brightness	−0.17	0.30 *	1.00				
Weight	−0.71 ***	0.41 **	0.52 ***	1.00			
Length	−0.25	0.04	0.18	0.42 **	1.00		
Thickness	−0.64 ***	0.02	0.59 ***	0.81 ***	0.55 ***	1.00	
Width	−0.56 ***	0.38 **	0.55 ***	0.84 ***	−0.05	0.65 ***	1.00

* *p* ≤ 0.05, ** *p* ≤ 0.01, *** *p* ≤ 0.001.

**Table 3 foods-09-00783-t003:** Protein ash and β-glucan contents (%) of three fractions among barley cultivars tested.

Cultivar	Dehulled Flour (DF)			Pearled Flour (PF)			Bran		
	Protein	Ash	β-glucan	Protein	Ash	β-glucan	Protein	Ash	β-glucan
ANDA	13.48 ^d^	1.61 ^bcd^	3.94 ^ef^	11.69 ^ef^	1.10 ^abc^	4.23 ^fg^	23.65 ^e^	4.87 ^bcd^	4.09 ^bc^
NIKI	11.39 ^f^	1.22 ^g^	3.87 ^f^	9.98 ^h^	0.72 ^f^	4.29 ^f^	21.39 ^f^	3.94 ^f^	3.58 ^fg^
PENI	12.31 ^e^	1.33 ^efg^	3.88 ^ef^	10.78 ^g^	0.96 ^de^	4.39 ^ef^	21.71 ^f^	4.07 ^f^	3.49 ^g^
DIMA	9.60 ^g^	1.27 ^fg^	4.29 ^cd^	8.51 ^i^	0.54 ^g^	4.54 ^de^	18.32 ^g^	4.69 ^de^	3.85 ^de^
CONS	16.06 ^a^	1.59 ^bcd^	4.67 ^b^	14.58 ^a^	0.91 ^e^	4.85 ^bc^	27.21 ^a^	4.73 ^de^	4.26 ^b^
MAKA	15.69 ^ab^	1.48 ^cf^	3.96 ^ef^	13.91 ^ab^	0.89 ^e^	4.64 ^cd^	26.87 ^ab^	4.51 ^e^	3.68 ^efg^
THKI	13.71 ^d^	1.61 ^bcd^	4.16 ^de^	12.14 ^de^	0.87 ^e^	3.99 ^gh^	26.00 ^c^	4.60 ^de^	3.75 ^ef^
SIOS	13.21 ^d^	1.73 ^ab^	5.08 ^a^	12.15 ^de^	0.69 ^f^	5.18 ^a^	23.94 ^de^	4.73 ^de^	4.78 ^a^
KOS	15.21 ^bc^	1.87 ^a^	4.57 ^bc^	13.47 ^bc^	1.09 ^abc^	4.83 ^bc^	25.76 ^c.^	4.87 ^bcd^	4.00 ^cd^
TIOS	11.90 ^ef^	1.64 ^bc^	4.27 ^d^	10.61 ^gh^	1.00 ^cde^	4.48 ^def^	21.84 ^f^	5.13 ^bc^	2.99 ^h^
THMI	13.51 ^d^	1.55 ^be^	3.41 ^g^	11.76 ^ef^	1.08 ^bcd^	3.75 ^h^	24.51 ^d^	4.79 ^cde^	3.00 ^h^
BIZA	12.17 ^e^	1.61 ^bcd^	3.91 ^ef^	11.08 ^fg^	0.88 ^e^	4.25 ^f^	23.82 ^de^	5.17 ^b^	2.98 ^h^
CYPS	15.18 ^bc^	1.89 ^a^	4.76 ^b^	12.63 ^d^	1.08 ^abc^	4.90 ^b^	26.03 ^bc^	5.20 ^b^	4.65 ^a^
IPOS	14.99^c^	1.70 ^abc^	4.07 ^def^	12.80 ^cd^	1.21 ^a^	4.30 ^ef^	26.03 ^bc^	4.73 ^de^	3.60 ^fg^
HLES	15.34 ^bc^	1.40 ^dg^	5.25 ^a^	14.39 ^a^	1.16 ^a^	5.27 ^a^	25.42 ^c^	5.58 ^a^	3.00 ^h^
*Mean ± SD*	*13.58 ± 1.81*	*1.57 ± 0.20*	*4.27 ± 0.50*	*12.03 ± 1.66*	*0.95 ± 0.19*	*4.53 ± 0.42*	*24.17 ± 2.42*	*4.77 ± 0.42*	*3.71 ± 0.57*

SD, standard deviation. Means in each column followed by a common letter are not statistically different (*p* > 0.05).

**Table 4 foods-09-00783-t004:** Phenolic acids, flavanols, tocotrienols, tocopherols and tocols contents of three fractions among barley cultivars tested as determined by HPLC methods.

Fraction	Cultivar	Phenolic Acids	Flavanols	Tocotrienols	Tocopherols	Tocols
		mg/100 g		μg/g		
Dehulled flour (DF)	ANDA	47.8 ± 0.8 ^de^	51.6 ± 2.7 ^e^	41.2 ± 0.6 ^de^	10.4 ± 0.6 ^cd^	51.6 ± 0.8 ^def^
	NIKI	40.1 ± 0.7 ^ef^	39.6 ± 1.4 ^f^	44.9 ± 1.0 ^cd^	8.3 ± 0.1 ^e^	53.2 ± 4.8 ^de^
	PENI	55.1 ± 0.4 ^a-d^	61.9 ± 3.0 ^abc^	53.9 ± 2.1 ^ab^	11.3 ± 0.1 ^c^	65.2 ± 2.0 ^ab^
	DIMA	48.8 ± 7.4 ^d^	54.0 ± 2.1 ^de^	45.2 ± 0.4 ^cd^	7.6 ± 0.4 ^e^	52.8 ± 3.2 ^de^
	CONS	51.2 ± 1.0 ^cd^	42.1 ± 1.6 ^f^	25.0 ± 0.5 ^f^	13.3 ± 0.5 ^b^	38.3 ± 1.3 ^gh^
	MAKA	60.4 ± 6.8 ^ab^	60.5 ± 0.8 ^bcd^	29.0 ± 0.2 ^f^	8.9 ± 0.3 ^e^	37.9 ± 1.9 ^gh^
	THKI	62.6 ± 3.0 ^a^	63.4 ± 5.9 ^ab^	47.9 ± 0.2 ^bc^	8.9 ± 0.3 ^e^	56.8 ± 0.2 ^cd^
	SIOS	52.6 ± 0.7 ^bcd^	54.5 ± 0.3 ^de^	40.6 ± 1.4 ^de^	8.4 ± 0.2 ^e^	49.0 ± 1.5 ^ef^
	KOS	58.8 ± 7.3 ^abc^	55.5 ± 7.9 ^cde^	36.7 ± 0.1 ^e^	8.2 ± 0.1 ^e^	44.9 ± 0.9 ^fg^
	TIOS	53.2 ± 2.0 ^bcd^	54.1 ± 0.2 ^de^	53.8 ± 0.6 ^ab^	8.5 ± 0.5 ^e^	62.3 ± 0.1 ^bc^
	THMI	34.1 ± 0.6 ^f^	58.7 ± 1.5 ^bcd^	54.5 ± 0.1 ^a^	14.2 ± 0.9 ^b^	68.7 ± 0.4 ^ab^
	BIZA	59.2 ± 1.0 ^abc^	61.5 ± 2.1 ^abc^	46.5 ± 1.7 ^cd^	7.9 ± 0.5 ^e^	54.5 ± 1.9 ^de^
	CYPS	60.1 ± 2.7 ^ab^	63.9 ± 0.1 ^ab^	40.6 ± 1.5 ^d^	11.0 ± 1.0 ^c^	51.6 ± 2.0 ^def^
	IPOS	58.6 ± 4.5 ^abc^	67.2 ± 1.1 ^a^	55.1 ± 3.6 ^a^	16.6 ± 0.6 ^a^	71.7 ± 3.9 ^a^
	HLES	58.2 ± 2.9 ^abc^	39.2 ± 4.5 ^f^	27.7 ± 2.1 ^f^	8.7 ± 0.3 ^e^	36.3 ± 2.3 ^h^
	*Mean*	*53.4 ± 8.2*	*55.2 ± 8.8*	*42.8 ± 9.8*	*10.1 ± 2.6*	*53.0 ± 10.8*
Pearled flour (PF)	ANDA	20.0 ± 0.3 ^e-h^	52.9 ± 1.3 ^cd^	26.7 ± 0.2 ^cd^	10.5 ± 0.3 ^a^	37.3 ± 0.1 ^b^
	NIKI	26.4 ± 2.5 ^bcd^	37.2 ± 0.4 ^f^	19.8 ± 0.2 ^fg^	4.5 ± 0.1 ^hij^	24.3 ± 0.3 ^ef^
	PENI	29.2 ± 0.8 ^bc^	74.5 ± 1.5 ^a^	35.6 ± 0.4 ^b^	7.5 ± 0.2 ^c^	43.2 ± 0.5 ^a^
	DIMA	36.2 ± 3.2 ^a^	56.5 ± 6.1 ^bc^	15.9 ± 1.2 ^h^	3.8 ± 0.2 ^jk^	19.7 ± 1.2 ^g^
	CONS	19.5 ± 0.8 ^fgh^	46.5 ± 2.1 ^e^	9.1 ± 0.5 ^i^	4.7 ± 0.3 ^ghi^	13.8 ± 0.2 ^h^
	MAKA	20.7 ± 0.7 ^e-h^	56.0 ± 0.4 ^bc^	15.8 ± 0.6 ^h^	4.7 ± 0.3 ^i^	20.5 ± 0.8 ^g^
	THKI	38.7 ± 4.1 ^a^	56.2 ± 1.8 ^bc^	24.1 ± 1.1 ^de^	4.7 ± 0.4 ^i^	28.1 ± 0.9 ^cd^
	SIOS	17.3 ± 0.4 ^h^	53.2 ± 0.1 ^cd^	18.3 ± 0.2 ^gh^	4.8 ± 0.1 ^fgh^	23.1 ± 0.3 ^efg^
	KOS	23.1 ± 1.5 ^d-g^	56.9 ± 3.9 ^bc^	23.2 ± 0.8 ^e^	5.4 ± 0.1 ^efg^	28.6 ± 0.8 ^cd^
	TIOS	24.7 ± 0.1 ^cde^	54.3 ± 5.8 ^c^	35.6 ± 2.4 ^b^	3.8 ± 0.2 ^jk^	39.4 ± 2.5 ^b^
	THMI	18.8 ± 0.4 ^gh^	45.1 ± 2.7 ^e^	40.5 ± 1.1 ^a^	5.6 ± 0.0 ^e^	46.1 ± 1.0 ^a^
	BIZA	24.25 ± 1.1 ^c-f^	62.0 ± 1.4 ^b^	22.7 ± 1.2 ^ef^	3.6 ± 0.2 ^k^	26.3 ± 1.4 ^de^
	CYPS	24.0 ± 4.0 ^def^	44.2 ± 5.3 ^e^	24.7 ± 1.1 ^de^	6.6 ± 0.2 ^d^	31.4 ± 1.2 ^c^
	IPOS	31.2 ± 5.2 ^b^	47.2 ± 0.4 ^de^	28.2 ± 2.1 ^c^	9.3 ± 0.3 ^b^	37.5 ± 2.2 ^b^
	HLES	20.5 ± 0.1 ^e-h^	54.0 ± 0.7 ^c^	15.7 ± 0.9 ^h^	5.5 ± 0.2 ^ef^	21.2 ± 0.9 ^fg^
	*Mean*	25.0 ± 6.4	*53.1 ± 8.7*	*26.2 ± 8.6*	*5.6 ± 2.0*	*29.4 ± 9.3*
Bran	ANDA	132.7 ± 4.1 ^e^	164.5 ± 6,5 ^bc^	152.1 ± 6.1 ^cd^	26.0 ± 2.8 ^bcd^	178.1 ± 3.3 ^cd^
	NIKI	136.3 ± 1.4 ^de^	124.6 ± 0,6 ^ef^	176.9 ± 6.9 ^b^	25.6 ± 1.0 ^b-e^	202.5 ± 7.9 ^b^
	PENI	159.4 ± 9.0 ^ab^	197.2 ± 5,0 ^a^	159.0 ± 5.0 ^bc^	25.5 ± 0.5 ^b-e^	184.5 ± 4.5 ^bc^
	DIMA	135.5 ± 2.1 ^de^	149.5 ± 0,7 ^cd^	210.8 ± 10.8 ^a^	21.4 ± 0.6 ^fg^	232.2 ± 10.1 ^a^
	CONS	150.3 ± 2.4 ^bc^	93.2 ± 2.8 ^g^	124.7 ± 4.7 ^ef^	22.9 ± 0.9 ^def^	147.6 ± 5.6 ^ef^
	MAKA	102.5 ± 4.0 ^g^	114.9 ± 5.7 ^ef^	131.2 ± 4.3 ^ef^	23.5 ± 0.5 ^c-f^	155.2 ± 4.8 ^e^
	THKI	139.4 ± 1.8 ^cde^	120.1 ± 7.0 ^ef^	204.4 ± 7.6 ^a^	22.3 ± 0.9 ^ef^	226.8 ± 8.4 ^a^
	SIOS	118.0 ± 1.8 ^f^	133.3 ± 1.0 ^de^	163.7 ± 13.7 ^bc^	29.9 ± 0.1 ^a^	193.5 ± 13.5 ^bc^
	KOS	112.0 ± 7.6 ^fg^	132.0 ± 3.0 ^de^	131.3 ± 4.7 ^ef^	26.4 ± 0.6 ^bc^	157.7 ± 5.3 ^de^
	TIOS	163.6 ± 9.6 ^a^	155.1 ± 8.3 ^c^	217.6 ± 7.6 ^a^	29.8 ± 0.6 ^a^	247.4 ± 7.0 ^a^
	THMI	109.5 ± 5.1 ^fg^	110.5 ± 3.5 ^fg^	206.1 ± 5.2 ^a^	30.9 ± 1.1 ^a^	237.0 ± 6.3 ^a^
	BIZA	140.8 ± 3.8 ^cde^	175.6 ± 9.0 ^b^	160.2 ± 2.9 ^bc^	22.9 ± 1.3 ^def^	183.1 ± 4.2 ^bc^
	CYPS	150.6 ± 0.5 ^bc^	208.0 ± 12.7 ^a^	132.3 ± 3.3 ^de^	27.7 ± 1.3 ^ab^	160.0 ± 4.6 ^de^
	IPOS	165.8 ± 5.5 ^a^	214.0 ± 24.0 ^a^	112.0 ± 2.9 ^f^	18.4 ± 0.5 ^g^	130.4 ± 3.4 ^f^
	HLES	146.3 ± 0.1 ^cd^	149.1 ± 11.1 ^cd^	124.9 ± 5.1 ^ef^	28.7 ± 1.3 ^ab^	153.7 ± 6.3 ^e^
	*Mean*	*137.5 ± 19.6*	*149.4 ± 36.1*	*160.5*± *35.0*	*25.5 ± 3.6*	186.0 ± 36.1

Means in each column followed by a common letter are not statistically different (*p* > 0.05).

**Table 5 foods-09-00783-t005:** Correlation coefficients among phytochemical components and antioxidant activity of barley cultivars.

	TPC	TFC	TPAC	PAs	Flavanols	Tocols	ABTS	FRAP
TPC	1.00							
TFC	0.97 ***	1.00						
TPAC	0.91 ***	0.85 ***	1.00					
PAs	0.95 ***	0.90 ***	0.92 ***	1.00				
Flavanols	0.88 ***	0.82 ***	0.93 ***	0.91 ***	1.00			
Tocols	0.92 ***	0.88 ***	0.86 ***	0.91 ***	0.83 ***	1.00		
ABTS	0.98 ***	0.97 ***	0.90 ***	0.95 ***	0.89 ***	0.93 ***	1.00	
FRAP	0.98 ***	0.97 ***	0.88 ***	0.95 ***	0.86 ***	0.93 ***	0.99 ***	1.00

*** Significant at *p* < 0.001. TPC, total phenolic content; TFC, total flavonoid content; TPAC, total proanthocyanidins; PAs, phenolic acids.

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
