# Peer review of "Bioactive Components and Antioxidant Activity Distribution in Pearling Fractions of Different Greek Barley Cultivars"

_foods, 2020, doi:10.3390/foods9060783_

Round 1
Reviewer 1 Report
This research paper is very well planned and prepared. Only citation for phenolic extraction procedure is needed. Discussion is enough and conclusions are adequate to the assumed research objectives.
The paper entitled: Bioactive Components and Antioxidant Activity Distribution in Pearling Fractions of Different Greek Barley Cultivars Authors: Maria Irakli *, Athina Lazaridou, Ioannis Mylonas, Costas G. Biliaderis is a valuable research study. It concerns current issues related to phytochemical profile of different fractions of barley cultivars. The topic is original and relevant, however, in my opinion, the interest to the readers may be limited due to barley cultivars endogenous only for Greek. Otherwise, very interesting is data concerning phenolic profile of free and bound phenolics. Authors should indicate why they only extracted twice the sample, and how many repetition of extracts they were prepared. In chapter 2.10. They should indcate which type of compounds they were analysed using HPLC and at what wavelength number. All the results are clearly presented and enough discussed. However, the Authors need to explain the differences between whiteness (how it was calculated) and lightness (brightness) in case of colour of barley cultivars and fractions. The conclusions are consistent with the aim of the study.
Author Response
Point 1. This research paper is very well planned and prepared. Only citation for phenolic extraction procedure is needed. Discussion is enough and conclusions are adequate to the assumed research objectives.
Response 1: The authors would like to thank reviewer for the complimentary comments. A relevant reference has been added.
Point 2. The paper entitled: Bioactive Components and Antioxidant Activity Distribution in Pearling Fractions of Different Greek Barley Cultivars Authors: Maria Irakli *, Athina Lazaridou, Ioannis Mylonas, Costas G. Biliaderis is a valuable research study. It concerns current issues related to phytochemical profile of different fractions of barley cultivars. The topic is original and relevant, however, in my opinion, the interest to the readers may be limited due to barley cultivars endogenous only for Greek. Otherwise, very interesting is data concerning phenolic profile of free and bound phenolics.
Response 2: In this study, we employed barley cultivars developed in our Institute via an on-going cereal breeding program. The emphasis was to explore the nutritional and phytochemical characteristics of Greek barley varieties in order to be used for food applications to enhance human nutrition. It would be certainly useful to evaluate foreign barley varieties adopted to the Greek environment in future studies. However, the findings in this paper concerning the differences in β-glucan and phenolic compounds content, and antioxidant capacity among the different milling fractions are applicable to all barley varieties independently from their growing location.
Point 3. Authors should indicate why they only extracted twice the sample, and how many repetition of extracts they were prepared.
Response 3: All barley samples were pearled two times and each measurement was repeated twice. We have mentioned that “Hulled barley samples (200 g in duplicate) were pearled” and also “Free phenolic compounds were extracted two times” . In the section of “statistical analysis” we have also added the sentence “All the characterization measurements were performed in four-fold”.
Point 4. In chapter 2.10. They should indicate which type of compounds they were analysed using HPLC and at what wavelength number.
Response 4: We have added a respective phrase according to the reviewer’s comment.
Point 5. All the results are clearly presented and enough discussed. However, the Authors need to explain the differences between whiteness (how it was calculated) and lightness (brightness) in case of colour of barley cultivars and fractions. The conclusions are consistent with the aim of the study.
Response 5: We have added we a respective sentence in the revised version of the manuscript.
Reviewer 2 Report
The manuscript was writen very carefully. Authors used lots of methods to determine the content of bioactive components and antioxidant activity; not only qualitative but quantative too. The results are clear presents, maybe the figure 2 is a bit small and therefore not fully legible.
Main question addressed by researchers is about distribution and quantification of nutritional and bioactive components in pearled fraction in fifteen Greek barley cultivars. Additionally they want to evaluate the antioxidant activity of their fraction to find out what kind of fraction/cultivar is the most valuable for human. This is very important questions because consumens should know what kind of food is the best for them from the nutritional point of view. Authors investigated fifteen cultivars what is helpful for comparing the right variety, which can then be propagated to farmers.
I have not essential remarks - in my opinion it is very good study.
Author Response
Point 1. The manuscript was writen very carefully. Authors used lots of methods to determine the content of bioactive components and antioxidant activity; not only qualitative but quantative too. The results are clear presents, maybe the figure 2 is a bit small and therefore not fully legible.
Response 1: The authors would like to thank reviewer for the complimentary comments. The Fig. 2 was separated into two figures according to reviewer’s comment in order to be legible.
Point 2. Main question addressed by researchers is about distribution and quantification of nutritional and bioactive components in pearled fraction in fifteen Greek barley cultivars. Additionally they want to evaluate the antioxidant activity of their fraction to find out what kind of fraction/cultivar is the most valuable for human. This is very important questions because consumens should know what kind of food is the best for them from the nutritional point of view. Authors investigated fifteen cultivars what is helpful for comparing the right variety, which can then be propagated to farmers. I have not essential remarks - in my opinion it is very good study.
Response 2: The authors would like to thank reviewer for the complimentary comments.
Reviewer 3 Report
The authors in this study demonstrate the phenolic characterisation and associated nutritional properties of barley barley cultivars in different pearling fractions.
There are several studies in literature that have characterised barley phenolics in other countries - variation observed between cultivars, GXE etc.
I would recommend using MS to identify specific compounds with associated antioxidant activity. This would strengthen the analysis.
Overall a well presented manuscript. But minor spelling and grammatical errors exist.
Author Response
Point 1. The authors in this study demonstrate the phenolic characterisation and associated nutritional properties of barley barley cultivars in different pearling fractions. There are several studies in literature that have characterised barley phenolics in other countries - variation observed between cultivars, GXE etc. I would recommend using MS to identify specific compounds with associated antioxidant activity. This would strengthen the analysis. Overall a well presented manuscript. But minor spelling and grammatical errors exist.
Response 2. Τhere are undoubtedly several studies in the literature concerning the nutritional and phytochemical characterization of barley cultivars, including different pearling fractions. However, in this study we’ve focused on Greek barley varieties in order to assist farmers in selection of superior barley cultivars for food uses based on evaluation of their antioxidant potential. The use of emerging analytical techniques, such as HPLC- MS analysis, would certainly strengthen the merit of the manuscript, and it will be considered is our future studies.
Reviewer 4 Report
It is a well-written manuscript. The only concern that I have is the choice of tables, charts to present the data. The tables provide a better overview of various cultivars clearly compared to the bar charts. The bar charts should include error bars, instead of letters showing the statistical differences. Most of the data, when looked closely show not much variation among cultivars and even though the variations could be statistically significant, the practical means of such variations have to be explained. Similarly, I would also like to see the comparison of barley with other major grains that will help the readers understand the importance of barley.
The manuscript has a wealth of data but the presentation should be improved and the choice of table suits this type of article than bar charts.
Author Response
Point 1. It is a well-written manuscript. The only concern that I have is the choice of tables, charts to present the data. The tables provide a better overview of various cultivars clearly compared to the bar charts. The bar charts should include error bars, instead of letters showing the statistical differences. Most of the data, when looked closely show not much variation among cultivars and even though the variations could be statistically significant, the practical means of such variations have to be explained. Similarly, I would also like to see the comparison of barley with other major grains that will help the readers understand the importance of barley. The manuscript has a wealth of data but the presentation should be improved and the choice of table suits this type of article than bar charts.
Response 1. We agree with the reviewer, but we have decided to present the results using both figures and tables. In our opinion, figures are more useful for readers to easily appreciate differences among samples, thus we adopted this format of data presentation for most of the research findings. For example, the Fig 2 was separated into two individual figures in order to be more legible. However, when they were not applicable (being more complicated), we rely on the use of Tables. In any case, standard deviations are shown in both tables and figures using error bars.